# Ectoplastic Art Therapy as a Genre of Contemporary Art

**Peter Tzanev**

Department of Art Psychology and Art Education, National Academy of Arts, 1000 Sofia, Bulgaria; peter_tzanev@yahoo.com

**Abstract:** Art therapy is the successor of "psychological Modernism", which during the late 19th and early 20th centuries included medical psychology as well as theories and practices related to more speculative practices of hypnosis, somnambulism, interpretation of dreams, automatic writing and spiritualism. Art therapy emerged in the second half of the 20th century as a new psychological genre and, the author argues, a new kind of art that offered the opportunity for psychological "salvation" in a "psychological society". This article explores an experimental project called "Ectoplastic Art Therapy" begun in 2002 by the author as a form of therapy and as a form of contemporary art. This therapy has been performed in various institutional settings, such as therapeutic centers, museums and galleries, as well as educational seminars and courses. Focusing on the usual marginalization that accompanies conventional art therapy within the established framework of the contemporary art system, this article examines the situations in which an art therapist could present his practice as a contemporary artist. The author prompts questions concerning the possible kinds of self-presentation that can be found in art therapy as a form of contemporary art.

**Keywords:** contemporary art; psychological Modernism; Ectoplastic art therapy; performativity

## 1. Introduction: Art Therapy and the Psychological Salvation of Art

The practice of what we now call art therapy began in the 1940s. This was contemporaneous with the failed revolution of the psychological Modernism or what Bramble (2015, p. 134), in his history of modernist occult practices, describes as, "The downfall of the modernist culture of soul".[1] Modernism was the first global movement in the history of art that clearly showed that art is a place of conflict between different modes of consciousness, or different patterns of mental life that seek their representations in art. The productive interaction between art and new psychological sciences became the most direct and exciting way to uncover the deepest expressions of the modern individual personality. At the same time, Modernism arose in a psychological context that was not monolithic. The literary scholar Joshua Gang claims that during the first decades of the twentieth century, there was competition between an array of psychological theories—including behaviorism, structuralism, and psychoanalysis—circulating simultaneously and competing against each other and that in this situation, to choose one psychological theory was to choose not only a model of mind, but also a set of aesthetic, philosophical, and political entailments (Gang 2013, p. 117). Of particular interest to me in developing my theory of Ectoplastic Art Therapy was the fact that medical psychology between 1880 and 1940 included theories and practices related to hypnosis, somnambulism, psychical research, dream interpretation, mediumistic psychology, automatic writing, faith healing, and spiritualism. According

---

[1] The epitaphic phrase "The downfall of the modernist culture of the soul" is the title of a chapter in John Bramble's book *Modernism and the Occult*, where he argues that syncretic occultism is the cornerstone of modernism.

to the historian Micale (2004, p. 18), psychological and aesthetic Modernism can be regarded as cultural domains with two parallel and competing discourses of the idea of The Modern and with shared metaphors for the dissolution of the real, visualizing the unseen, and the primitive in the modern. In a similar plan, the historian Corinna Treitel in her book *A Science for the Soul: Occultism and the Genesis of the German Modern* uses the term psychological Modernism when explores the ways in which occult practices from the 1870s to 1945 had an effect on German modernism. According to Treitel, the occult movement belonged to the larger spectrum of German attempts to elaborate "alternative modernities", and the distinctive feature of this movement was its embrace of a primarily "psychological solution" to the problems of modernity (Treitel 2004, p. 51). Treitel claims that the modern occult sciences and occult texts assumed canonical status within the field of psychological Modernism (Treitel 2004, p. 72). The category psychological Modernism describes the complex existence of a separate historical phenomenon in the period of the late nineteenth century and the first half of the twentieth century, during which heterogeneous psychological attitudes and ideas coexisted. It can be said that art therapy is the strange and surprising successor to this phenomenon, as it continues to develop the art of psychological Modernism in the second half of the twentieth century and the early twenty-first century (Nikitin and Tsanev 2018).[2]

The triumph of psychology as an independent modern discipline opened new territories to the development of Western individualism. Psychology not only legitimized a new kind of autonomous subjective space based on psychological terms and concepts but also offered new models for its representation. It would not be an exaggeration to say that many of the representational models of Symbolist art were based on the categories of Transcendent psychology,[3] and that Abstract art was born in the secret world of Theosophical psychology,[4] and almost all ideas of Surrealism were related to Psychoanalysis, just as Gestalt psychology looked behind the discoveries of Op art, and Abstract expressionism was immersed in the archetypal messages of Jungian psychology, that Minimalism was involved in Phenomenological psychology, and Conceptual art was in coalition with the successful invasion of Cognitive psychology. The claims of psychology to be a new science of "Man" were oriented towards transforming the thinking and desires of modern humanity into completely psychological phenomena. While it can be very difficult to demonstrate how different psychological constructs are realized in art, it is much easier to see the function of art in societies with developed psychological technologies and the influence of these technologies on the historical development of art. The emergence and development of art therapy as a distinct psychological genre after the 1940s demonstrated the role of art in the newly "psychological society" (Tsanev 2014).

What is art therapy from the point of view of art? What is the meaning of the invention of art therapy for art? The emergence of the phenomenon of art therapy reveals two new important features of art. First, art therapy declared the independent existence of a new form of art with purely psychological functions. Secondly, art therapy went beyond demonstrating the connection between art and psychology, which experimental aesthetics and Art Psychology had been doing for 150 years. The purpose of art therapy was, instead, to create a new kind of connection between art and personality.

---

[2] In my part of the book *Image and Consciousness in Art Therapy*, I explore how the origin of art therapy was tied to the expectations of the avant-garde movements of the twentieth century to turn modern art into the most important part of the human psychic evolution. Psychological Modernism was related to the idea of higher consciousness, and in this context, it saw art as a psychological tool for the development of that higher consciousness (Nikitin and Tsanev 2018).

[3] Transcendent psychology of Carl du Prel was very popular in the late nineteenth century. Du Prel's Transcendent psychology made the unconscious and its occult phenomena the site where the scientific concerns met spiritual ones. Du Prel understands human consciousness as Janus-faced. One face inhabited the waking world of everyday sense experience while the other lived in the unconscious world of dream, trance, clairvoyant vision, and telepathic suggestion. Between the two lay a barrier of awareness that normally blocked one face from knowing the other. See (Treitel 2004, p. 42).

[4] Russian occultist Helena Blavatsky, the chief founder of the Theosophical movement, defined Theosophical psychology as the science of the soul. She postulated that every mental change is signalized by a molecular change in the brain substance. For Materialism, mental changes are caused by molecular changes, and for Spiritualism (believers in a soul) molecular changes are caused by mental changes.

Art therapy as a form of art arose with the intention to psychological liberate modern humanity from the restrictive adherence to materialism and physiology. Since art therapy interacts with a subject that is absorbed in a psychological reality, we can assume that art therapy treats art as an opportunity for psychological salvation of the individual in "psychological society".

## 2. Why Did Art Therapy Not Become a Genre of Art?

Art is an extension of the place that individuals inhabit but also extends the structure of human space and behavior through the new artworks that function as artistic research objects. The contradictions of the new modern art objects, such as the "collage", the "readymade", and the "surrealistic object" that appeared in the early 20th century, formed a new niche in the individual's space, which can be described as "perceptual extension". Psychological views of art regard art objects not as something absolutely objective but as a process of interaction in which one can only be induced to knowledge when there is a situation that puts him in a state of uncertainty. In this sense, art is an instrument that achieves a particular psychological goal set by the individual.

The scientific claims of occult psychological Modernism on the study of unknown natural laws and hidden forces in the human mind have had a powerful influence on avant-garde artists who constructed projects around the psychological dimensions of invisibility. The most obvious evidence of this influence can be found in the impact that the book *Thought-Forms*, published in 1901 by the British theosophists Anni Besant and Charles Leadbeater, left in the work of artists such as Wassily Kandinsky and Hilma af Klint. The illustrations in *Thought-Forms* of different types of "psychic auras" and "spiritual vibrations" can be considered some of the first experimental images in which a new abstract aesthetics of psychological Modernism was manifested. The new theosophical psychology captured their imaginations with the idea that it offers a path to the essence of true art. Theosophical art was addressed to the psychological subject and focused on the psychology of what appears outside of the senses. Modernism was focused on an imaginary model of humanity and on a theurgic concept of art, according to which the artist has special mental abilities, allowing them to overcome the limitations of perception and making them able to reveal the true essence of what is hidden. In 1910, the esoteric philosopher Rudolf Steiner stated that there are so many imperfect works of art, because the artist cannot know what appears outside his perceptions (Bowlt 1996, p. 51). According to John Bowlt, as a response to this dilemma, Russian modernists turned to the idea of a knowledge that is beyond perception by using synesthesia, trance and altered states of consciousness. In Petersburg, during the first two decades of the 20th century, there was a group of avant-garde artists inspired by a panpsychic model of art. This group of artists, which researchers identify with the term the "organic school" (Wünsche 2015), was a unique in that it viewed the experimental figure of the artist as an opportunity to expand humanity's sensory abilities. Examples include Jan Ciągliński's concept of "integrative vision", or the theoretical psychological views of the military doctor, theorist, patron and artist Nikolai Kulbin and his "psycho-impressionism", Elena Guro's "anti-urbanism" and, above all, about Michael Matyushin's psychological concepts and his method of "extended vision". The circle of artists around the charismatic figure of Kulbin was the first to articulate the idea of "psychological art" (Howard 2016). In 1908, Kulbin founded and funded the "Art-Psychological Group—Triangle", the first organization of artists with consciously declared psychological intentions.

Artists from the organic school, regardless of their success and influence, did not invent a new genre of psychological art. Even later, when in Kandinsky, Malevich, and Matyushin created art practices that worked like a psychological laboratory, they still had no intention of creating a purely psychological genre of art.

During the first decades of the twentieth century, images were central to developing theories of the unconscious as well as a spiritualist understanding of the universe. In 1914, Albert von Schrenck-Notzing published in Munich the book *Phenomena of Materialisation* which included disturbing

images of mediums with ectoplasm,[5] an unusual form of psychological fantasy. In the same year, Carl Gustav Jung began his *The Red Book*, which opened the way for the future development of art therapy.

Schrenck-Notzing's interest in ectoplasm co-developed with that of French physiologist Charles Richet, who coined the term in 1894. It was only after Richet won the Nobel Prize in 1913 that Schrenck-Notzing published his book. Ectoplasmic processes, Schrenck-Notzing believed, had their origin in the unconscious mind of the entranced medium in terms of materialized dream-images, that is, ephemeral, externalized precipitates from the medium's psychical impressions, imagination, and memories.

*The Red Book* (1914–30) is a collection of Jung's personal writings and drawings, chronicling the explorations of his own unconscious. With *The Red Book*, Jung discovered a new psychological genre that transformed psychotherapy into a practice concerned with art making in the context of personal expression and the higher development of the personality.

The photographs included in the *Phenomena of Materialisation* that were supposed to prove the existence of ectoplasm had discovered an obvious fraud. Schrenck-Notzing admitted that mediums deceptively smuggled pins into the séance room and replicated the ectoplasm using a comb, gauze and a handkerchief. Jung never published *The Red Book* during his lifetime. He feared that a book filled with his own dreams and hallucinogenic drawings could ruin his career.

Schrenck-Notzing and Jung were choosing contradictory psychological phenomena and they turned them into even more contradictory psychological experiments, which they thought of as new therapeutic tools.

Contemporaneous with these occult image practices was the emergence of new genres in art making. The invention of new genres led to a profound change in the psychological function of art. These new genres created an experimental situation in which the individual was exposed to unusual conditions and left to identify their place in relation to the institutional framework of art. The explosion of new genres, which began when Picasso invented a collage in 1912, not only made a contribution to a particular medium but also provided a new model of artistic behavior. The twentieth century witnessed an unprecedented proliferation of new genres (Table 1).

**Table 1.** Table showing the dominant attitude towards the art medium and the most important new genres in the main historical periods of art for the past 120 years with moments related to the idea of how art therapy can influence the significance of the artistic experience. Table created by author P. Tzanev.

| Modern Art | Postmodern Art | Contemporary Art |
|---|---|---|
| 1900–1960 | 1960–1989 | 1989–2019 |
| Pure medium | Intermedia/Multimedia | Medium transgression |
| **New genres** | **New genres** | **New genres** |
| Collage | Installation | Post-medium art |
| Abstraction | Appropriation | Artistic research |
| Readymade | Institutional critique | New genres public art |
| 1942—Adrian Hill coined the phrase | 1972—Roger Cardinal coined the term | 2013—Massimiliano Gioni includes Carl Jung's |
| Art Therapy | Outsider Art | Red Book (1914–30) at the 55th Venice Biennale |

---

5    The term ectoplasm describes the white immaterial emanation that surrounds a person in a deep state of trance. As described by the investigators, the ectoplasm moved in an autonomous animate manner, responding to touch. The substance was reported to develop into rudimentary limb-like forms, often assuming a two-dimensional form first. Occasionally, full human limbs and heads were reported to form, appearing lifelike and responsive to the environment. Finally, the forms would either gradually dissolve, with the ectoplasm being reabsorbed by the medium, or suddenly vanish into thin air (Sommer 2012).

The dramatic increase in new artistic genres was a product of both new practices and new attitudes and according to David Galenson, not only did many artists want to do new things, but they often wanted to underscore the novelty of these new things by giving them new names (Galenson 2009, p. 132).

According to German art historian Hans Belting, art becomes the sphere of the artist, who assumes control of the image as proof of his or her art and the crisis of the old image and the emergence of the new concept of art are interdependent (Belting 1994, p. 16). If we develop Belting's claim that "images live in their media much as we live in our bodies" (Belting 2005, p. 306), we can say that modern images live best in pure media and postmodern images prefer to live in intermedia or multimedia, and contemporary images live in a condition of medium transgression.

A very important part of the life of the images in their preferred media is the specific niche of their existence, which is made possible by the invention and establishment of each new genre in art. It is not possible to imagine modern art and the impact of its images without the emergence of such new genres as collage, abstraction, and readymade. Postmodern art also imposed its images in new genres such as appropriation, installation art, and institutional critique. Contemporary art is now experimenting with images that offer possible organization of the visible across new genres such as post-media art and art research.

From the very first moments of its origin, art therapy had failed to find a niche in existing art genres. For example, Kandinsky commented on the healing properties of colors and the transformative potential of abstract images in the metaphysical context of the spiritual tasks but did not insist on separating these functions into a new artistic genre. Art therapy as a practice continued to exist for a long time in an unclear genre condition. Even the famous art therapy of Jackson Pollock from late 1938 to September 1940 with Jungian analyst Dr. Joseph Henderson (Rothenberg 2017) did not have a separate genre status in the painter's biography from his other paintings and drawings. Similarly, Dubuffet coined the term "art brut" in 1948 for art produced by intense psychological urgency outside aesthetic norms, rather as a cultural phenomenon than as a psychological genre. The distinction between the makers of Art Brut and the work of Art Informal artists such as Dubuffet is a blurry one.

The psychological society has expanded psychological expectations in relation to the object of art. A work of art expands its complexity through a new type of identification, which is increasingly based on psychological characteristics. Thus, art becomes involved in thinking about the hypothetical psychological subjects that accompany the art object in its appearance to us. The art of the psychological epoch inherited from its magical, religious, and metaphysical past a special feeling that high art can unfold its reflexivity only in the realm of truth, which is not subject to verification by any external reasons and bases. At the same time, in the contemporary era, art cannot continue to be based on hidden theology. Art must replace magical superstition, religious faith and metaphysical transcendence with a confidence in a world entirely filled with psychological forms of reflexivity. In this sense, for a psychological society, art does not function only as an extension of experience (which was a project of modernism) or only as a critique of experience (which was a project of the avant-garde). In the late modernist society, art is primarily a psychological event that aims, on the one hand, to permanently overcome aesthetics by overcoming the metaphysical nature of speculative aesthetic experience, and, on the other hand, to permanently overcome art by overcoming the concept of art. In both cases, this overcoming occurs by understanding that no one can be the absolute master of the psychological.

In the second half of the twentieth century, art therapy largely developed as an anachronistic experimental field of psychological research of the human personality, which had little direct interaction with the radical achievements of neo-avant-gardist art and its anti-subjective pathos. In the 1960s and 1970s, art introduced and legitimated the subject of art, in which the expressive illusion of psychological interiority was rejected by the instrumentalism of conceptual objects. Influenced by post-structuralist psychoanalysis, and with the development of discursive psychology at the end of the 20th century, global conceptualism found its stable balance between the dualistic tensions of a non-expressive subject and the decentralized traumatic nature of deconstructive art.

### 3. Art Therapy as an Art Strategy in the Context of Four Historical Periods in the Development of Art Therapy

We are used to thinking about the development of art therapy around two main poles: the first is "art psychotherapy" evolving from psychiatry and the second is "art as therapy" emerging from art education (Junge 2010, pp. 34–35). Therefore, it is very difficult define the place of art therapy in the art world.

When art therapy appears in the art world, it does so through the figure of the artist, whose approach tends to fall along one of the following four lines: (1) the artist as a "psychological hostage"; (2) the artist as a "psychological rebel"; (3) the artist as a "psychological consultant"; (4) the artist as a "psychological researcher". Interestingly, these four trends coincide with the four historical periods in the development of art therapy as a profession from the 1940s to the present, and which I described below.[6]

The first period of art therapy (from the late 1930s to the end of the 1950s) was associated with an increased interest of artists in psychoanalysis. These artists believed in the expressive power of art and the natural potential of the art of healing. Art therapy was important for artists as a transformative experience. Art therapy was dominated by a cult of the inner world and the unconscious. Carl Jung's theories were influential for many artists to harness the imagery of their dreams and to experiment with techniques that freed the artist from conscious control over their creativity. During this period in art therapy, the figure of the artist appeared as a "psychological hostage". By the term psychological hostage, I mean that the artist was initially admitted to psychotherapy only as a patient and a victim. When from 1939 to 1940, in New York City, Jackson Pollock was a patient of the Jungian psychologist Dr. Joseph Henderson, Pollock submitted 83 drawings to Henderson as material for analytical aids in therapy (Cernuschi 1992). Significantly, the drawings seem to have been created at the same time to explain the analysis and as an important part of the evolution of Pollock's original artistic process. As American art critic Kuspit (1995, pp. 121–35) points out, Pollock's equal psychological involvement in psychotherapy and the psychoanalytic sensibility of his drawings remain misunderstood. Two years after the Pollock's therapy sessions, the British artist Adrian Hill coined the term "art therapy" and began his art therapy work, which was documented in his book "Art Versus Illness". During the same time, in New York City, Margaret Naumburg introduced art as a therapeutic practice by calling her work "dynamically oriented art therapy" (Junge 2010, p. 5).

The second period (from the early 1960s to the late 1970s) was connected with the establishment of art therapy as a psychotherapeutic practice and independent profession. Artists were interested in the ideas of anti-psychiatry and humanistic psychology. Art therapy considered art as a kind of authentic self-communication. The artists involved in deconstructing the idea of psychopathological imagery and the inclusion of "psychiatric art" to the modern art canon. During this period in art

---

[6]  The commonly accepted periodization of the history of art therapy takes three main stages: the classical period (1940s to 1970s), the middle years (1970s to mid-1980s), and contemporary art therapy (mid-1980s to present). This periodization, made by American art therapist Randy Vick, is based on the evolution of the theoretical literature in the field of art therapy (Vick 2003, pp. 5–15). British art therapist Chris Wood divides the development of art therapy into three historical periods: from the late 1930s to the end of the 1950s; from the early 1960s to the late 1970s; and from the early 1980s to the mid-1990s (Wood 1997, p. 145). The main difference between these two approaches to the history of art therapy is that Wood's periodization contains an earlier and diffuse first period, followed by two more clearly defined second and third periods. Some authors believe that there is a clearly defined fourth period after 2000 associated with the interaction of art therapy with neuroscience and positive psychology (Tsanev 2014). According to Wood, each period is related to a certain tendency: during the first period, when the first ideas about using art as therapy in hospital settings began to emerge, attention was focused mainly on the expressive power of art, relying on its natural healing potential; the second period passes under the flags of anti-psychiatry movement, the ideas of humanistic psychology and the differentiation of art therapy as an independent profession; the third period is in the wake of increasing professionalization of art therapy and its ever closer attachment to the psychotherapeutic theory and practice. Such a development of art therapy reveals how gradually art began to play a subordinate role to the goals of psychotherapy. The conflict that consistently arose between art and the practice of art therapy was the real danger that artistic interpretation could replace the interpretations of psychotherapy. In art therapy, there are many approaches that can be distinguished by their belonging to a particular psychological school, but the most essential criterion remains the way in which each art therapy determines itself in terms of art.

therapy, the figure of the artist appeared as a "psychological rebel". By the term psychological rebel, I am referring to the involvement of many artists in the field of art therapy who have been enthralled by the romantic aspects of the anti-psychiatric movement. This is especially true in the UK. A feature of the British Psychoanalytic School is the dynamic exchange of ideas and approaches to the psychology of art conducted by psychoanalysts such as Melanie Klein, Ronald Fairbairn, Hannah Segal, Wilfred Bion, Donald Winnicott and Marion Milner, and art thinkers, such as Adrian Stokes, Ernst Gombrich, Anton Ehrenzweig, Peter Fuller and Richard Wollheim. The intense dialogue between psychoanalysis and art was spread in three directions. The first direction was a clinical model that has developed and established art therapy as an independent paramedical profession. The second direction has helped the development of psychoanalytically oriented art criticism and art theory. The third direction was related to the integration of psychoanalysis into the models of art education. The British Association of Art Therapists was founded in 1964 and The American Art Therapy Association was founded in 1969.

The third period (during the 1980s and 1990s) is connected to increased professionalization of art therapy and increased interest of artists to the techniques and institutions of psychotherapy. The academization of art therapy as an educational project redirected the artists toward professional psychotherapeutic training. The emphasis on the creative process shifted from the notion of free sublimation to the notion of the discursive strategies. During this period in art therapy, figure of the artist appeared as a "psychological consultant". With the term psychological consultant, I mean that the expectations to the artist in art therapy are limited to the professional development of art techniques, and not to experimentation with new psychological methods and approaches.

The fourth period (after 2000) has witnessed the increased interest of artists in the impact of neuroscience on the development of the therapeutic disciplines involved in the phenomenology of human perception and behavior. The emergence of positive psychology, neuroaesthetics and neuroarthistory has changed the role of art therapy as an incubator of new ideas about mental health. The return of art therapy as contemporary art in controversial rhetorical field of critical psychology started with the unusual alliance of art therapy with neuroscience and new genres of psychological art. During this period in art therapy, the figure of the artist appears as a "psychological researcher". This fourth historical period in the development of art therapy coincided with an increased interest in performative practices in the field of visual arts, which created opportunities for some forms of art therapy to be represented as equal participants in the contemporary art system. This moment is important because it marks the transition of the artist's place in the field of art therapy from a "psychological hostage" and a passive object of psychological study to a "psychological researcher" with a high degree of self-reflective abilities.

## 4. Art Therapy as a Performative Artwork

Within the field of visual arts, "performativity" is a concept linked to the situatedness of the viewer in relation to the artwork (Petersen 2015, p. 243). According to the German art historian Dorothea von Hantelmann, known for her critique of the concept of performativity in the visual arts, what the notion of the performative in relation to art actually points to is a shift from what an artwork depicts and represents to the effects and experiences that it produces (Von Hantelmann 2014). The pioneer of performance studies Richard Schechner argues that accepting the performative as a category of theory makes it increasingly difficult to sustain a distinction between appearances and reality, facts and make-believe, surfaces and depths.[7]

As Anne Ring Petersen (2015, p. 243) writes, the concept of performativity denotes a production or interactivity that the subject does not have full control over; an encounter with the work as an

---

[7] Schechner found seven functions of the performance as seven interlocking spheres: to entertain, to create beauty, to mark or change identity, to make or foster community, to heal, to teach or persuade, to deal with the sacred and the demonic (Schechner 2013, p. 46).

event that befalls the subject and in which the object "does something" to the subject. In the project "Ectoplastic art therapy", the performativity has been construed as a situation that include modes of interactivity and intersubjective exchange at the level of imaginary communicative acts. The concept of the performative in art recognizes the productive dimensions of art works and makes it possible to understand art therapy as a mode of operation where subjectivity is formed through the performative act itself. In an effort to integrate art therapy and fine art, what I have dubbed "Ectoplastic art therapy" was staged from the beginning in different art world contexts, such as exhibitions at art museums and galleries, open discussions at artist-run spaces, and seminars at universities (Figure 1).

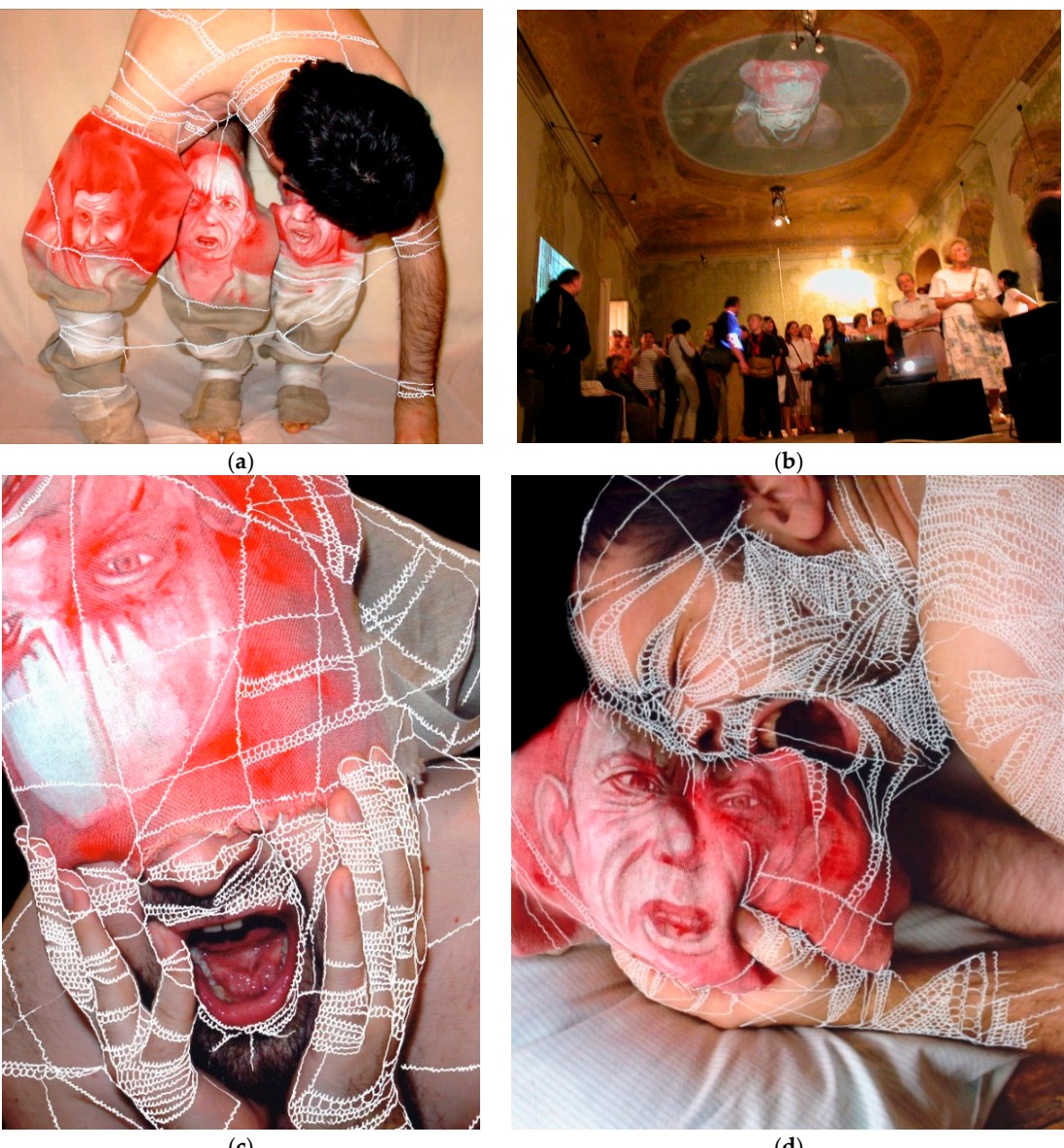

**Figure 1.** Peter Tzanev, Ectoplastic art therapy in different formats: (**a**) Hallucinating objects, intermedia, 2003. Club for Discussions "Close Together", Hambara, Sofia, 12 February 2004; (**b**) Hallucinating objects, intermedia, National Art Gallery, Sofia, 9 September 2005; (**c**) Hallucinating objects, intermedia, Project "Nigredo", Central Prison of Sofia, 2005; (**d**) Hallucinating objects, intermedia, donumenta 2005, Regensburg; photos: P. Tzanev.

Ectoplastic art therapy uses white ink on the photographed surfaces. The photograph itself is an intermediate tool that serves as a platform for a complex drawing process. On the one hand,

the painted surface risks turning into a decoration, becoming itself, and on the other hand, if spatial norms are observed, drawing becomes an indistinguishable part of the photo itself and is again hidden in it. Any drawing turns the physical reality into an imaginary one, and the purpose of drawing with ectoplastic art therapy is different. Ectoplastic art therapy uses physical reality, which was already an imaginary reality. Thus, by drawing on this imaginary reality, the participant in ectoplastic art therapy creates drawings that eliminate the illusion of drawing because they allow him to perform actions simultaneously in physical and in imaginary reality (Figure 2).

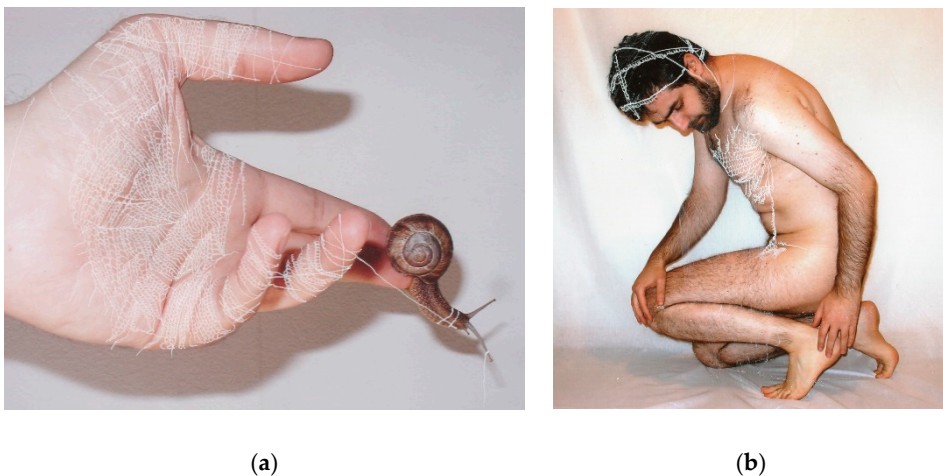

(**a**)                                         (**b**)

**Figure 2.** Peter Tzanev, Ectoplastic exercises: (**a**) Non-sensitive drawings (Snail), 2007; (**b**) Ectoplastic drawing, 2007; Images created by author.

Initially, the ectoplastic technique came out of my own art practice, inspired by the over-drawing on photographs by the Austrian artist Arnulf Rainer, together with my self-limiting conceptual doctrine, declared in 2002, to work using only white. During this period, the overdrawing technique was tested with children in the form of what I intended to be an interactive art therapy.

Back in the late 1990s, I discovered, during my work with young children, some peculiar features of their spontaneous reaction when giving them the opportunity to paint on photographs and reproductions of paintings and drawings. I had noticed that there was an interesting relationship between their mental state and the way they prefer to interact with the symbolic or formal characteristics of the images. Especially in my work with children with different mental health problems, I was intrigued by the ability of scribbles and drawings to perform functions that were not intended to represent and symbolize. In this regard, the theory of the Belgian philosopher Lenain (1997) about monkey art as a disruptive play was very important to me. For Lenain, the scribbles and markings of primates were interventions that never occurred regardless of the structure of the destructive operations on a given surface. The covering of a sheet of paper was a systematic disruption of the pictorial field and each marked trace over figure represented an annihilation of form. I have found that for some children, scribbles on photographs and illustrations of paintings and drawings could not reach phases in which they attempted to imitate, repeat, or add entirely new configurations as simple abstract figures or representational drawings which interact with movements and spatial structures of the images.

In 1998, the English neuropsychologist Humphrey (1998), a specialist in the evolution of human intelligence and consciousness, proposed an interesting hypothesis about the similarity between iconic Upper Palaeolithic images and the unusual drawings of a three-year-old child with autism (the famous Nadia). Humphrey had noted the remarkable similarities between the cave paintings and Nadia's drawings, in which the figures overlap without regard to the presence of the former. Autistic children are unusually attentive to detail in a sensory array while being relatively uninfluenced by—and even may be unaware of—the larger context. Their tendency to focus on parts rather than wholes helps them to ignore the context and see through it. Autistics perform exceptionally well on the so-called

'hidden figure' test, where the task is to find a target figure that has been deliberately camouflaged by surrounding lines. Experiments with 'embedded figures' for visual attention and visual search have revealed superior performance in individuals with autism. The detection of specific details embedded in an overall picture is the result of overcoming the stimulus of the whole pattern and deficits in the socioemotional perception.

In 1999, I successfully defended my Ph.D. thesis *Universal Motifs in Art through the View of Children's Drawings*, where I had a separate chapter on psychopathological imagery. Around 2001, I was involved in continuous art therapy with an autistic boy. Gradually, based on the atypical interaction of this autistic boy with faces, such as depicting all faces in a strange stereotypical way, the idea of overdrawing on faces as well as overdrawing on photos of his own face had crystallized in me. It was then that I discovered the special nature of white over-drawings.

A very important aspect of this new form of art therapy is that the drawing is done using white rather than black outlines. This very important finding that drawing with white has the potential to reduce the neglect of the existing visual context has had a very great effect on the further development of the technique.

Drawing images on existing images is an activity that creates a special kind of images. An overdrawing image is an image that cannot be separated from the context of the visual surface on which it is made. The degree of isolation and, consequently, the immersion of that special kind of images in the visual reality of the surface on which they are created is directly related to the therapeutic aspects of drawing as a process. The perception and creation of such images unlock capacities to cultivate heightened body consciousness and extend the potential for unblocking stereotypic habits.

The first public presentation of this technique was at my solo exhibition entitled "Victims of Decoration" at the legendary XXL Gallery in Sofia in November 2002. At the end of the same year, I published my first book, *Art, Psychopathological Imagery and Children's Drawings* (Tzanev 2002), which contains the first publication in Bulgaria and proposes a method of art therapy for children with autism.

After 2005, I began to use the word "ectoplastic" to describe my therapeutic work, because of the formal similarities to ectoplasmic photography from the early twentieth century. The great success of the Metropolitan Museum of Art exhibit on the subject in 2005, "The Perfect Medium: Photography and the Occult", with its provocative ambiguity, had a compelling effect on me. Thus, I replaced my references to "hallucinatory drawings" and "hallucinatory objects" with "ectoplastic drawings" and "ectoplastic objects". The turning point was the successful participation in numerous exhibitions, as well as the positive psychological transformation that accompanied the development of the new art therapeutic method.

Ectoplastic art therapy has its origins as a self-therapeutic experiment, but over the years, it has been tested on different groups of people. This approach has been particularly successful in individuals with post-traumatic stress disorder (PTSD), major depressive episode (MDE) and autism spectrum disorder (ASD).

In parallel with my art therapy practice with children and adults, I began to present and share my findings on the therapeutic aspects of "ectoplastic communication" with images in various exhibitions and art forums.

Ectoplastic psychological sessions with the public are conducted in two ways. The first way is closer to the art therapy rituals and offers the opportunity for volunteer participants to paint with white ink on their own photos that were brought by them. The sessions are based on direct psychological interpretation that leads to unexpected and effective psychotherapeutic insights. The second way is indirect and uses the epistemological potential of ectoplastic images. In this second way, I rely on the provocative nature of the ectoplastic images I create. My performative approach is close to what the German art historian Horst Bredekamp (2018) calls "Image Acts", in which the images not only reproduce or transmit, but also actively produce their own knowledge.

Images are entangled in a network of other images, imaginations and memories and cannot be reduced to information or messages. For example, my early ectoplastic experiments, based on

interactions with photographs of the mentally ill, which I discovered in Russian psychiatry textbooks published in the 1950s, provoked contradictory reactions in the places where they were exposed. The ectoplastic images were discussed in 2004 with an audience interested in the relationship between psychology and art, and in 2005, they were projected onto the ceiling of the small hall in the National Art Gallery in Sofia and on the walls at the Central Male Prison in Sofia. Absolutely all reactions were directly provoked by the way the images had affected the viewers. The discussions with the audience were entirely devoted to issues related to psychology, mental illness and art therapy.

With the method of Ectoplastic art therapy, the body, personality and consciousness are considered from the very beginning as a surface. Thus, thoughts, ideas, and momentary states are also present as surfaces. They are transparent surfaces on which we can leave marks and even make some kind of delicate pattern on them. Ectoplastic art therapy implies an imaginary interdependence between the normal ways we perceive our visible body as a host of images and the transfer way of the experience of our body through the overdrawing images, which resembles the bodily presence in front of a mirror. The use of simulated mental states adjusts a reflective surface and the interactions with the images it produces. The complex process of exhibiting artworks in different formats and art media further enhances the potential for self-communication in which every part of the self-reflections represents re-negotations of a new identity. For example, when an art therapist traces his behavior as an opportunity for a certain kind of pictorial interaction, he is forced to regulate his self-image with the act of drawing that has already taken place in his imagination. This is an example of a very delicate interactive process in which interaction and direct perception become the basis for intersubjectivity. (Figure 3). Ectoplastic art therapy investigates the complex self-therapeutic situation in which a psychotherapist falls into the role of artist and an artist into the role of psychotherapist in the hyper-psychological medium of art therapy. During the long process of art therapy, our body acts as a psychologization 'machine'.

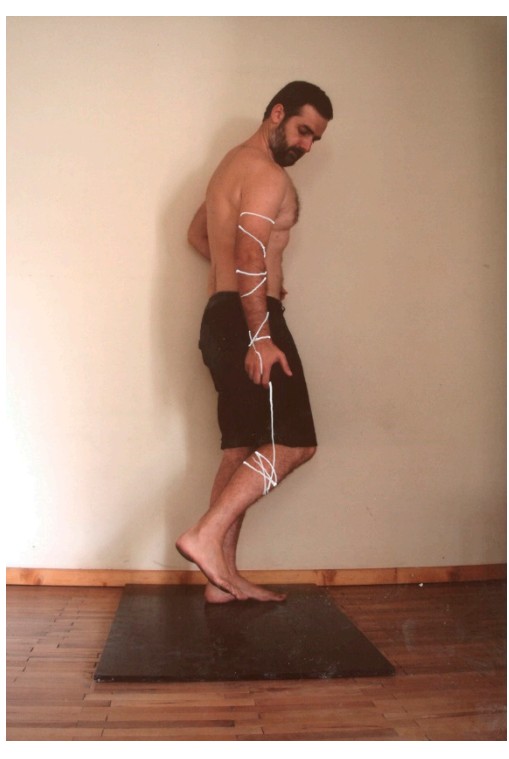 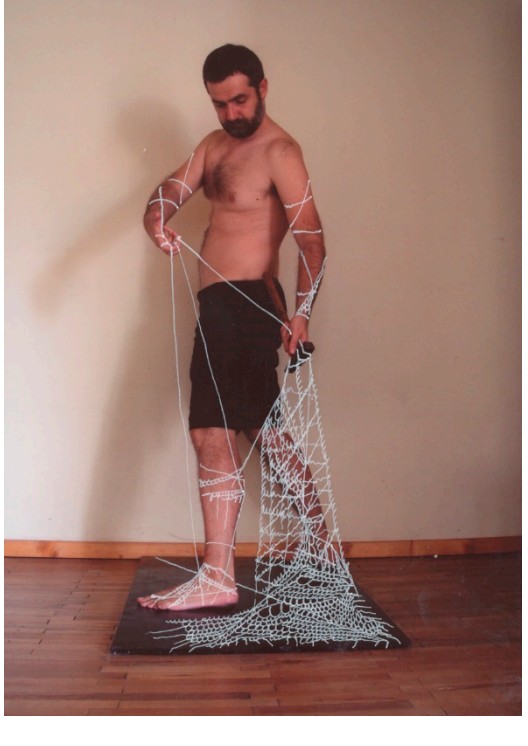

(**a**)             (**b**)

**Figure 3.** Peter Tzanev, Ectoplastic exercises: (**a**) Ectoplastic drawing, 2007; (**b**) Ectoplastic drawing, 2007; Images created by author.

With the help of objects of art, we realize that we exist in formations immersed in a complex mixture of non-classical reflection entities, such as instinct, language, power, and art. The subject in the context of ectoplastic art therapy is considered not as a self-sufficient entity, but as a site for transferring images. In this sense, we can say that our ectoplastic twin is not an imaginary subject, but an imaginary psychological place for the transforming power of images. By adding white ink ectoplastic marks on a photo or placing white objects and parts of white objects (Figure 4) in real spaces, we perform ritual actions that "remove transparency" of the real world (Figure 5).

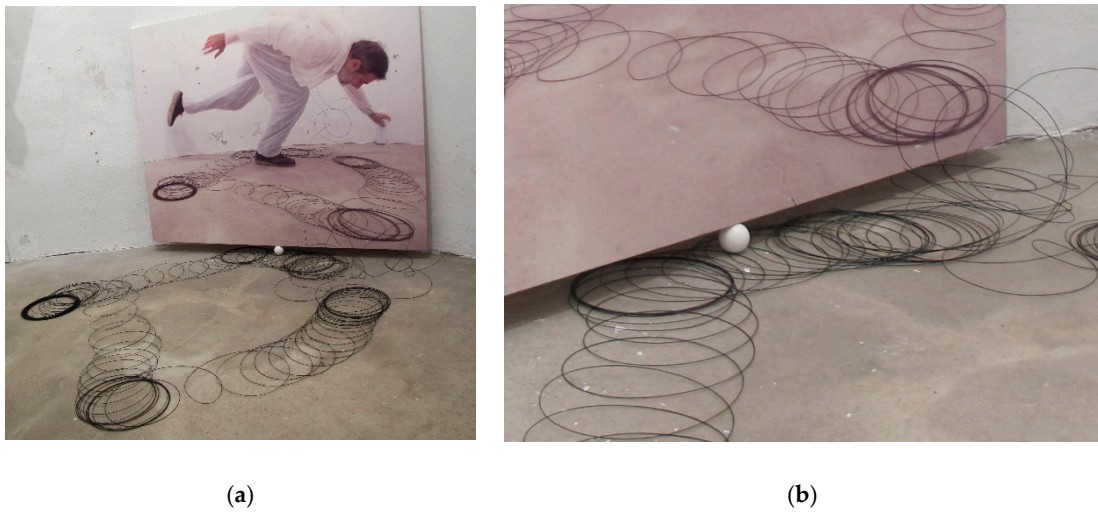

(**a**)                    (**b**)

**Figure 4.** Peter Tzanev, Ectoplastic exercises: (**a**) Out-of-Body Experience, 2018; (**b**) Out-of-Body Experience, (detail), 2018; photos: P. Tzanev, 2018.

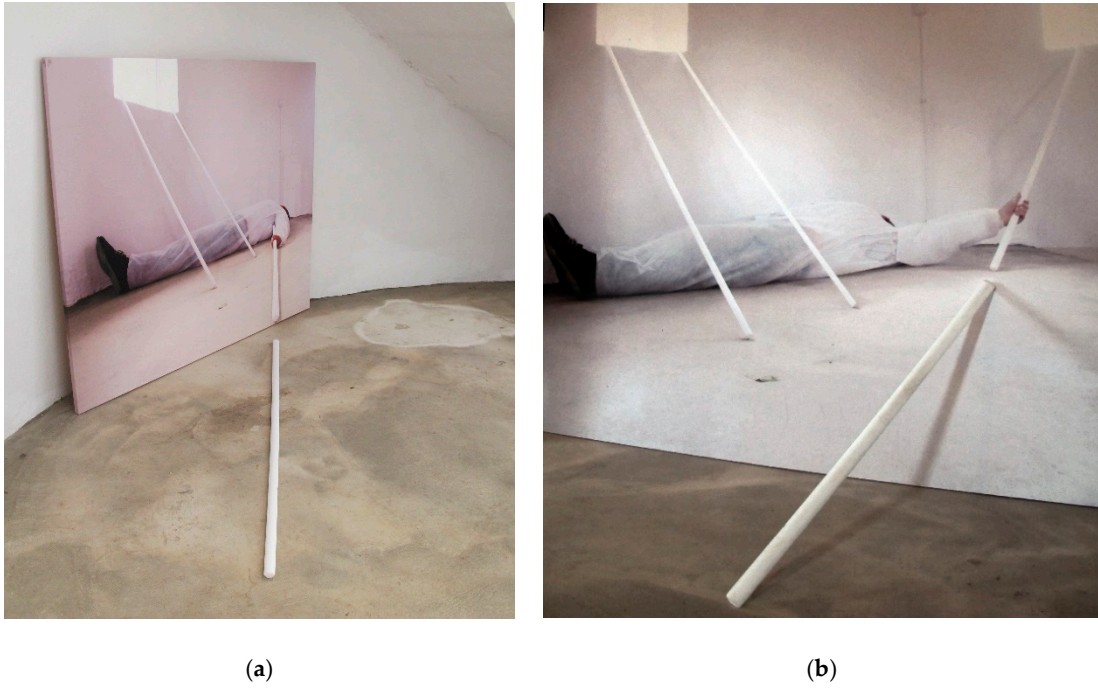

(**a**)                    (**b**)

**Figure 5.** Peter Tzanev, Ectoplastic exercises: (**a**) Out-of-Body Experience, 2018; (**b**) Out-of-Body Experience, 2018; photos: P. Tzanev, 2018.

Ectoplastic art therapy regards the forms of consciousness in art as an access simultaneously to the formation of both perceptual and intellectual concepts. One of the most important goals of ectoplastic art therapy can be defined as the effort of drawing in space using tools that cannot be instrumentalized.

Objects of art capture the non-reflective content of images that affect the mind, but they are not directly provided to our attention.

## 5. Conclusions: Ectoplastic Art Therapy as a Genre of Art

Ectoplastic art therapy is possible only in a dynamic framework, given by the experimental subject, which stands behind neo-avant-garde objects of art. In this dynamic framework, it is important to design artistic practices reviewing the avant-garde as reconnecting art with living life in historical networks, since the "artist is not just unlike others but absolutely different because of the concept of art" (Soussloff 1997, p. 5).

Art as a tool in art therapy is not a direct intervention on the disease, but indirect intervention on the concepts of the disease and the notions of treatment. In this context, art therapy can be defined as an activity that constructs a world beyond the dialectic of disease and healing. Thus, art therapy becomes an autonomous activity, which, with the means of psychology, creates imaginary doubles of both the disease and the treatment. But more importantly, art therapy attempts to overcome the world, constructed by existing cultural definitions of disease and treatment. In this sense, many contemporary forms of art therapy work towards the liberation of consciousness as a form of perception, both in medicine and in art.

Ectoplastic art therapy aims to overcome the traditional use of art in art therapy, linked to the dominant role of expressive art therapy models. Ectoplastic art therapy recommends the differentiation of art therapy as an independent field both outside art and outside psychology.

**Funding:** This research received no external funding.

**Acknowledgments:** Many thanks to the institutions discussed in this paper who have kindly provided support of my ideas.

**Conflicts of Interest:** The author declares no conflict of interest.

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
