# Peer review of "Ectoplastic Art Therapy as a Genre of Contemporary Art"

_arts_

Round 1
Reviewer 1 Report
Overall this is a very intellectual and interesting paper that situates art therapy within both the art world and the psychological world and history - we need many more such reflective historical and critical analyses of art therapy!
at the same time, there are gaps in the literature - the writer seems to come from the art world and maybe theosophical art therapy - there is a lack of literature of art therapy to justify the claims in the introduction ( ie, what are the 4 stages of art therapy? what literature backs up the assumptions stated? The author needs to read some basic art therapy books and situate them int the texts to prove his point
In the central paper, there is a jump to the examples as if they describe the whole of art therapy
if they exemplify what is said in the introduction, this has to be clearer, if not it has to be clearer why they are used -they are very untypical of art therapy art, and thus cannot be used as examples alone
Author Response
Response to Reviewer 1 Comments
Point 1: Overall this is a very intellectual and interesting paper that situates art therapy within both the art world and the psychological world and history - we need many more such reflective historical and critical analyses of art therapy! at the same time, there are gaps in the literature - the writer seems to come from the art world and maybe theosophical art therapy - there is a lack of literature of art therapy to justify the claims in the introduction ( ie, what are the 4 stages of art therapy? what literature backs up the assumptions stated? The author needs to read some basic art therapy books and situate them into the texts to prove his point.
Response 1: I am absolutely agree with the comment that my text contains gaps in literature on art therapy. I thought the Special Issue would be more oriented towards art theory, so I cut out that part of my text about art therapy history (for example the 4 historical stages of art therapy). Now, in the revised version, I add this part, revised according to the recommendations.
Point 2: In the central paper, there is a jump to the examples as if they describe the whole of art therapy if they exemplify what is said in the introduction, this has to be clearer, if not it has to be clearer why they are used - they are very untypical of art therapy art, and thus cannot be used as examples alone.
Response 2: In the revised version of my article, I develop a completely new separate part called "Art Therapy as an Art Strategy in the Context of Four Historical Periods in the Development of Art Therapy". This part is between the part that asks the basic question "Why did Art Therapy not become a Genre of Art?" and the next part that interprets "Art Therapy as a Performative Artwork". I hope that this revisions will justify my claims and why I use the untypical examples of art therapy art.
Reviewer 2 Report
This article considers art therapy, and in particular an experimental project 'Ectoplastic Art Therapy', as a form of contemporary art. It raises and addresses, original and relevant questions about the relationship between art therapy and contemporary art. It takes a refreshing perspective and opens up consideration of the relationship between these related, yet relatively unexplored, fields of practice. In doing so it contributes an interesting analysis and argument for how art therapy might be considered in relation to the field of contemporary art. The experimental project 'Ectoplastic Art Therapy' embodies a new practice to be considered in the histories and discourses of art therapy and contemporary art.
I found the article to be clearly written, structured and referenced. It is organised through the exploration of key questions and I believe it will be of interest to a wide audience. And I anticipate it will open up further discussions and consideration of this interconnectedness of contemporary art and art therapy from different perspective and practices.
Overall I find the article is rigorous and I consider it to be acceptable in its current form. There is one minor point on line 166 where the term 'legalised' is used. I think this may not be correct and possibly the term 'legitimated' might be more accurate to the meaning intended.
In terms of suggestions for development in future articles. I would have liked to hear more about the experience of the art practice of 'Ectoplastic Art Therapy'. To further unfold what the artist experiences when involved in this practice and to understand more about the origin of the term 'Ectoplastic' when in relationship the term 'art therapy'. The images are intriguing and I consider could bear more commentary in terms of context, content and experience. However I consider this to be a fascinating project and discussion and a valuable contribution.
Author Response
Response to Reviewer 2 Comments
Point 1: Overall I find the article is rigorous and I consider it to be acceptable in its current form. There is one minor point on line 166 where the term 'legalised' is used. I think this may not be correct and possibly the term 'legitimated' might be more accurate to the meaning intended.
Response 1: I am agree that on line 166 the term 'legalised' is used not correct and the term 'legitimated' is more accurate.
Point 2: In terms of suggestions for development in future articles. I would have liked to hear more about the experience of the art practice of 'Ectoplastic Art Therapy'. To further unfold what the artist experiences when involved in this practice and to understand more about the origin of the term 'Ectoplastic' when in relationship the term 'art therapy'.
Response 2: The project is in progress. I have a book and publications in Bulgarian and Russian. I'm currently preparing a book in English that introduces the experience of the art practice of 'Ectoplastic Art Therapy'.
Point 3: The images are intriguing and I consider could bear more commentary in terms of context, content and experience. However I consider this to be a fascinating project and discussion and a valuable contribution.
Response 3: Thank you very much for the positive review.
Round 2
Reviewer 1 Report
There are improvements to the paper overall I think it's worth publishing because it's original- although a little problematic in terms of it's duel focusAuthor Response
Response to Reviewer 1
There are improvements to the paper overall I think it's worth publishing because it's original- although a little problematic in terms of it's duel focus
Response: Thank you very much for the positive review.